# Post-competition recovery strategies in elite male soccer players. Effects on performance: A systematic review and meta-analysis

Albert Altarriba-Bartes[1,2☯], Javier Peña[1,2☯]*, Jordi Vicens-Bordas[2,3,4☯], Raimon Milà-Villaroel[5‡], Julio Calleja-González[6‡]

**1** Sport Performance Analysis Research Group (SPARG), University of Vic-Central University of Catalonia, Vic, Barcelona, Spain, **2** UVic-UCC Sport and Physical Activity Studies Centre (CEEAF), University of Vic-Central University of Catalonia, Vic, Barcelona, Spain, **3** Department of Medical Sciences, Research Group of Clinical Anatomy, Embryology and Neuroscience (NEOMA), School of Health and Sport Sciences (EUSES), University of Girona, Girona, Spain, **4** School of Health and Sport Sciences (EUSES), Universitat de Girona, Salt, Spain, **5** Global Research on Wellbeing (GRoW), Blanquerna School of Health Sciences-Ramon Llull University, Barcelona, Spain, **6** Department of Physical Education and Sports, Faculty of Education and Sport, University of the Basque Country, UPV/EHU, Vitoria-Gasteiz, Spain

☯ These authors contributed equally to this work.
‡ These authors also contributed equally to this work.
* javier.pena@uvic.cat

**Data Availability Statement:** All relevant data are within the manuscript and its Supporting Information files.

## Abstract

### Aims

The main aim of the present review was to update the available evidence on the value interest of post-competition recovery strategies in male professional or semi-professional soccer players to determine its effect on post-game performance outcomes, physiological markers, and wellness indicators.

### Methods

A structured search was carried out following the PRISMA guidelines using six online databases: Pubmed, Scopus, SPORTDiscus, Web of Science, CINAHL and Cochrane Central Register of Controlled Trials. The risk of bias was completed following the Cochrane Collaboration Guidelines. Meta-analyses of randomized controlled trials were conducted to determine the between and within-group effects of different recovery strategies on performance, physiological markers and wellness data. Final meta-analyses were performed using the random-effects model and pooled standardized mean differences (SMD).

### Results

Five randomized controlled trials that used Compression Garments (n = 3), Cold Water Immersion (n = 1), and acute Sleep Hygiene Strategy (n = 1) were included. Greater CMJ values at 48h for the intervention group (SMD = 0.70; 95% CI 0.14 to 1.25; p = 0.001; $I^2$ = 10.4%) were found. For the 20-m sprint and MVC, the results showed no difference either at 24h or 48h. For physiological markers (CK and CRP) and wellness data (DOMS), small to large SMD were present in favor of the intervention group both at 24h (-0.12 to -1.86) and

**Funding:** The authors received no specific funding for this work.

**Competing interests:** The authors have declared that no competing interests exist.

48h (-0.21 to -0.85). No heterogeneity was present, except for MVC at 24h ($I^2 = 90.4\%$; p = 0.0012) and CALF DOMS at 48h ($I^2 = 93.7\%$; p = 0.013).

## Conclusion

The use of recovery strategies offers significant positive effects only in jumping performance (CMJ), with no effects on the 20-m sprint or MVC. Also, the use of recovery strategies offers greater positive effects on muscle damage (physiological markers and wellness data), highlighting the importance of post-match recovery strategies in soccer.

## Introduction

The interaction between training load, fatigue, adaptation, and recovery is an element of extreme complexity comprising factors of a very different nature [1, 2]. According to the literature, maximizing the performance of an athlete is not only a matter of training, but it is also affected by a wide array of intrinsic and extrinsic elements [2, 3]. Current evidence highlights that enough and optimal recovery is necessary to prevent health problems and to achieve peak performance and the choice of recovery strategies by coaches and athletes may be crucial [4, 5]. Proper recovery strategies can lead athletes to better performances, helping them to feel more rested and healthy [6]. However, high-performance athletes face a wide array of daily training stimuli that may not allow complete recoveries [7], emphasizing the need for optimal recovery strategies based on individual fatigue thresholds [4, 8].

Recovering as quickly as possible, restoring pre-performance levels is considered a crucial element of success in almost every athletic discipline [9]. For this reason, coaches and athletes are always in a continuous search for the most effective strategies to speed up post-exercise recovery [2, 9–11]. However, precisely defining the concept of "recovery from exercise" is a challenging mission due to the number of variables affecting an optimal recovery [12]. This pioneering idea has inspired a multi-factorial approach to the "physiology of recovery," evidencing the need for more conclusive research [13].

Placing the focus on fatigue in elite competitive soccer, we observe that the average player at this level is exposed to high-congested game schedules with a mean of 60 competitive games played per season, equating 5.5 games per month [14] or one game every 4.3 days [15]. Consequently, a lot of physical and psychological stress is imposed on professional soccer players [9, 16]. Players participating in two games per week and less than or equal to four recovery days are under a substantial risk of sustaining an injury. It is estimated that it is more than six times higher, compared to having only one game per week and a recovery time of six days or more between competitions [17–19]. Imposing load without enough recovery might also be an essential factor leading to illnesses or injuries [20, 21].

Among other performance factors in soccer, repeated sprint ability, jumping ability, maximal strength seem to be reduced immediately after a game; and the time needed to recover from training sessions or competitive events fully may vary between 48 hours and 96 hours depending on the authors and the physical fitness values analyzed [18, 22–30]. Besides, biochemical markers in team sports are also altered inconsistently after training or competition, showing relevant differences in the recovery profile of every sport [31]. Particularly in soccer, CK and hormonal parameters seem the most relevant biomarkers of the recovery process [32].

Establishing the importance of recovery, several studies show non-significant differences in injury risk, running performances, or pace in technical activities during congested competitive

periods in professional soccer players [33, 34]. Soccer players seem to be able to cope with the physical demands of consecutive games [34–37]. Thus, the decline in performance can be attributed to an increase in game interruptions and not to the effect of physical fatigue, and it may be a common trend to overestimate fatigue-induced performance declines [35]. Player's covered distances and velocities also show dependency on contextual game factors such as the venue and the result of the game [36]. To experience transient residual fatigue over the games and the season is something common in professional soccer players, causing adverse effects on the on-field physical performance and predisposing to overuse and non-contact injuries [18, 37–40]. Minimizing the effects of travel fatigue should also be taken into account [41], given that traveling long distances by plane has a significant effect on the subjective ratings of jet-lag, neurological fatigue, and sleepiness [42].

The knowledge about physical performance profiles, players management (squad rotation) recovery strategies, and time courses seems to be an essential factor in getting a realistic approach to recovery, establishing an optimal periodization design for the season, and optimizing players' readiness for the upcoming competitions. [16, 33, 38, 43–46].

To enhance the recovery process, the more common strategies employed by athletes include ergogenic aids, hydrotherapy, active recovery, stretching, compression garments, and massage [47, 48]. These methods are frequently used by professional soccer players, being nutrition, sleep, compression garments, cold-water immersion, and contrast water therapy, the ones with a better subjective perception [49]. However, in many cases, scientific evidence is not taken into account before implementing these strategies, showing inadequacies of sports science knowledge translation to the day-to-day practice [4]. Abaïdia and Dupont [50] proposed a practical recovery protocol based on an extensive scientific revision, finding a high grade of recommendation for several nutritional strategies and hydration, cold water immersion, whole-body cryotherapy, and compression garments. In this proposal, other recovery strategies such as sleep, massage, foam rolling, electrical stimulation, and massage were considered inappropriate, or its benefits in physical performance and recovery were not clear. Moreover, other authors concluded that even active strategies were largely ineffective for improving post-exercise recovery, offered some benefits compared with passive ones [51, 52].

Specifically, in soccer, some studies show that active recovery neither has effects on neuro-muscular recovery nor in antioxidant response to competitive games and muscle soreness [27, 45, 53]. Others found it useful, reducing muscle pain, concluding that it may help to restore performance abilities such as vertical jump [54]. Cold-water immersion is another of the most common strategies employed and has been reported as effective improving muscular damage and discomfort and overall fatigue perception after training and competition, but not having a definite positive effect on physical performance [55–58]. Modern techniques, such as electro-stimulation and foam roller, have also shown a significant effect on the recovery in agility and perceived muscle soreness [59, 60] while compression garments have reduced histological damage in some experimental studies [61]. However, the studies with professional or semi-professional soccer players are scarce, and consequently, decision making very complex.

Several authors have tried to find pooled positive effects of using combinations of different recovery strategies. Kinugasa & Kilding [62] observed higher positive effects on perceived recovery after combining cold-water immersion and active recovery. In another study, whole-body vibration (WBV), in combination with a traditional cool-down reduced perceived muscle pain and enhanced recovery faster than the protocols without WBV after a soccer-specific drill [10]. Other authors have demonstrated that no recovery strategy is more effective than the others. However, the use of combined strategies tended to be more effective than a simple strategy [63]. To the best of our knowledge, no systematic review has analyzed the empiric use of these strategies in professional soccer settings previously.

Therefore, the main aim of the present study is to review the available evidence on the value of post-match recovery strategies and interventions in male professional or semi-professional soccer players in order to determine its effect on post-match performance outcomes, physiological markers, and wellness indicators.

## Materials and methods

### Design

A systematic review and meta-analysis focusing on the effects of different recovery strategies in professional soccer contexts were reported following the recommendations of the Preferred Reporting Items for Systematic Reviews and Meta-analyses statement (PRISMA) [64]. Before the search, a review protocol based on PRISMA-P [65] was completed (S1 File) and registered at PROSPERO (ID = CRD42018094854). The review protocol was updated during the review process and is available at http://www.crd.york.ac.uk/PROSPERO/display_record.asp?ID= CRD42018094854 (07 November 2019)

### Search strategy and study selection

A systematic computerized literature search was performed using six online databases: Medline (PubMed), Scopus, SPORTDiscus, WOS (Web of Science), CINAHL, and Cochrane Central Register of Controlled Trials (CENTRAL). The search included articles published before May 20[th], 2020. All databases were searched using Boolean operators with the following medical subject headings (MeSH) and free text words for critical concepts related to recovery and soccer performance: "Athletes," "Sport," "Recovery," "Match," "Performance," "Feeling perception." The eligibility of the studies was formulated according to the following PICOS criteria, which returned relevant articles in the field using a snowballing approach:

- Population: elite professional or semi-professional male football or soccer players.

- Intervention: structured interventions comparing methods and control groups.

- Comparison: studies that compare different recovery modalities or between a modality and control group.

- Outcomes: physical performance was taken into account as a primary outcome. Subjective perception, wellness, technical, tactical, and physiological performance were considered as secondary outcomes.

- Study design: randomized clinical trials were included.

Studies were included if 1) were randomized controlled trials (RCTs) with participants randomly separated into equal groups (control group and intervention group); 2) participants were semi-professional or professional adult football/soccer players; 3) recovery strategies were performed after a competition. Studies were excluded if: 1) female players were taken into account. Only full-text publications in English were considered.

The complete search strategy for each database can be found in the S2 File. The searches were customized to accommodate the layout and characteristics of each search tool. The reference sections of all identified articles were examined, and a hand-search of it was also conducted for other potentially relevant references.

One author selected papers for inclusion (AAB). Titles and abstracts obtained by the search were screened and downloaded into Mendeley Desktop (Glyph & Cog) for a subsequent full-text review. Cross-references and duplicates were removed. All publications potentially relevant for inclusion in the meta-analysis were independently assessed by two reviewers (AAB

and JVB). Any discrepancies at this stage were resolved during a consensus meeting, and a third (JP) reviewer was available if needed.

## Outcome variables

For the primary outcome, changes in muscle strength, sprint and jump performance values obtained from different tests after using recovery modalities were considered.

For the secondary outcomes, changes in psychological, wellness, and physiological data were considered.

## Data extraction

General study information, participants, intervention characteristics, and outcome measures were extracted independently by two reviewers (AAB and JVB) using a specific standardized data extraction form (S3 File). When studies provided insufficient data for inclusion in the meta-analysis, the first author of the study made contact with the corresponding author(s) to determine whether additional data could be provided; in other cases, data was extracted from graphs using Digitizeit digitizer software (https://www.digitizeit.de).

## Risk of bias

Methodological quality was not implemented, as no evidence for such appraisals and judgments exists and, therefore, can be confusing when interpreting results [66].

A bias is a systematic error, or deviation from the actual effect, in results or inferences. The authors assessed the risk of bias in RCTs following the Cochrane Collaboration's tool for assessing the risk of bias in randomized trials [67]. The items on the list were divided into six domains: selection bias (random sequence generation, allocation concealment); performance bias (blinding of participants and researchers); detection bias (blinding of outcome assessment); attrition bias (incomplete outcome data); reporting bias (selective reporting); and other bias. For each study, bias domain was judged by consensus (AAB and JVB), or third-party adjudication (JPL) and was characterized as "high" (a plausible bias that severely weakens confidence in the results); "low" (a plausible bias unlikely to seriously alter the results); or "unclear" (plausible bias that raises some doubt about the results). A quote from the study report, together with a justification for the judgment, was provided.

## Statistical analysis

Descriptive data of the participants' characteristics were reported as mean (SD). All meta-analyses calculations were conducted with the R software with meta and metafor packages for met-analysis (Version 3.5.1.). Descriptive analyses and figures of risk of bias were performed using Microsoft Excel for MAC, version 16.29.1 (Microsoft, USA). Mean and standardized mean differences (Hedges' g) and 95% CI for each group were calculated. The analysis of pooled data was conducted using a random-effect model [68] to estimate the change for each group at the same measurement time on primary and secondary outcomes. For the secondary meta-analysis, the mean difference between primary and secondary outcomes was collected to estimate the change from baseline to each time measurement for each group (control and experimental groups). Standardized mean differences were weighted by the inverse of the variance to calculate the size of the effect and 95% confidence interval. Cohen's criteria were used to interpret the magnitude of the effect: $<|0.50|$: small; $|0.50|$ to $|0.80|$: moderate; and $>|0.80|$: large [69]. Heterogeneity was assessed using Cochran's Q statistics and its corresponding p-value as well as the $I^2$ statistic, which describes the percentage of variability in effect estimates attributable to

heterogeneity rather than chance when $I^2$ was >30% (30–60% representing moderate heterogeneity) [66]. Publication bias was assessed with funnel plots and Begg's test. Significance was set at $p < 0.05$.

In the case of studies reporting recovery at different time frames such as 20h and 44h, those values were assimilated to the ones reported in the literature, 24h and 48h.

## Results

The initial search identified 4184 references (Fig 1). No other references were identified through the examination of reference lists and citations of relevant articles. After the identification of duplicates, 3402 titles and abstracts were screened. Seven studies remained for further full-text analysis. Subsequently, 2 studies were excluded. The reasons for exclusion were that participants were not football or soccer players; or data on primary outcomes (performance) was not assessed in the study. In the end, five studies were included in the final review process.

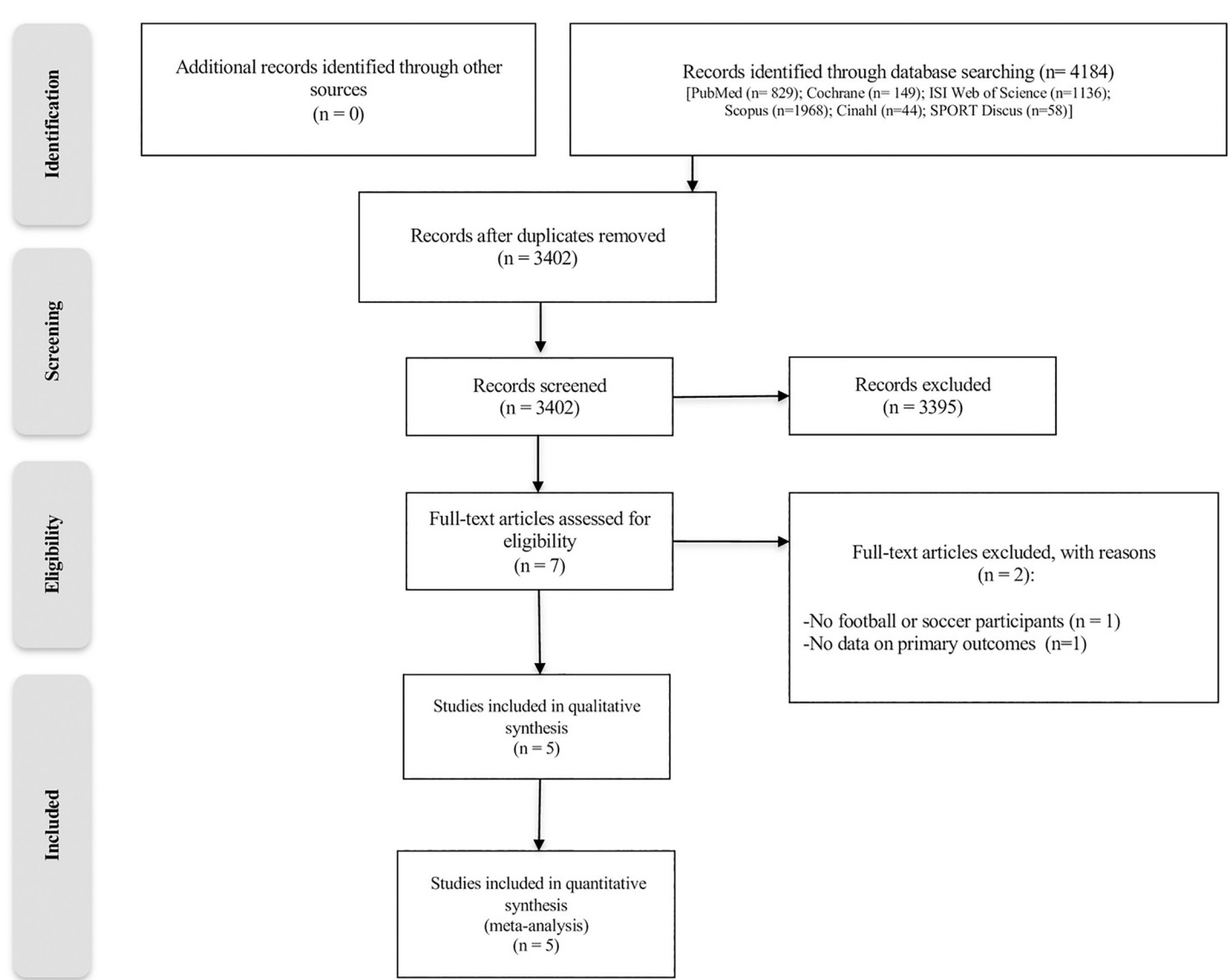

**Fig 1. Eligibility flow diagram showing the selection process for the inclusion studies in this meta-analysis.** n: sample size.

## Description of studies

Five RCTs [57, 70–73] were included in this review, with their most relevant characteristics being summarized in Table 1. A total of 69 participants were included in the review, with a mean age of 20.8 ± 1.3 years with a range of 18 to 28 years The competitive level of the soccer players in the studies was semi-professional [57, 71–73], and elite or professional [70]. From the included studies, three assessed the effects of wearing lower-body compression garments

**Table 1. Characteristics of the included randomised controlled trials.**

| Study | Population and level N (male); age ± SD | Intervention description | Group: Intervention and Control characteristics | Primary Outcome | Secondary outcomes | Results, conclusions and intervention effect |
|---|---|---|---|---|---|---|
| *Ascensão et al.* [57]. | 20 junior soccer players National team leagues IG (10); 18.1 ± 1.8 years CG (10); 18.3 ± 0.8 years | Effect of immediate post-exercise CWI single session on soccer players | After match for 10 minutes IG: CWI 10ºC CG: TWI 35ºC | SJ (cm) CMJ (cm) 20-m sprint (sec) MVIC (Kg) | DOMS Muscle damage: CK (U/L), Mb (µg/L) Inflammation CRP (mg/L) | Decrease in SJ at 24h and CMJ at 24h and 48h in the TWI group[b] Decrease in CMJ at 24h in the CWI group[b] Decreases in peak quadriceps strength in the TWI group at 24h and 48h and in CWI at 48h[b] Quadriceps strength greater at 24h in CWI group[a] CWI more effective than TWI at 24h for quadriceps and calf DOMS and at 30min for hip adductors[a] CK increased in both groups at 30min, 24h and 48h[b] and more in the TWI at 24h and 48h[a] Mb increased in both groups at 30min[b], more in the TWI[a] CRP concentrations increased in both groups at 30min and 24h[b], but again more in the TWI than in CWI[a] |
| *Clifford et al.* [70]. | 11 elite professional soccer players 19.0 ± 1.0 years | Effect of wearing lower body garments fitted with cooled phase changed material (PCM) on accelerating functional and perceived recovery after a game | 45 min after match for 3 hours. 5 mmHg IG: PCMcold 15ºC CG: PCMwarm 22ºC | CMJ (cm) MVIC (N) | BAM+ MS BFQ | MVIC at 36 h and 60 h post was greater with PCMcold than PCMwarm[a] MS post 36 h and 60 h was lower with PCMcold than PCMwarm[a] No differences in CMJ or BAM + between groups. PCMcold was more effective than the PCMwarm after the intervention according to BFQ[a] |
| *Fullagar et al.* [71]. | 20 highly trained semi-professional soccer players 25.5 ± 4.6 years | Effect of an acute sleep hygiene strategy (SHS) on physical and perceptual recovery of players after a late-night game. | IG: SHS lights dimmed, eye-masks and ear plugs, cool temperature rooms (~17˚C). No technological or light stimulation ~15–30 min prior to bedtime. 7h 30 min in bed. CG: NSHS allowed to use mobile phones and TV. 5 h 30 min in bed. | External load Internal load CMJ (cm) YYIR2 (m) | Objective and subjective sleep data General recovery state Sleep chronotype RPE Psychological recovery Physiological recovery Muscle damage: CK (mg/ml) and urea(mg/dl) Inflammation: CRP (mg/dl) | Greater sleep duration in SHS compared to NSHS on match night[a] Less sleep duration with NSHS[b]Greater wake episodes on match night for SHS[a] No differences between conditions for any physical performance or venous blood marker. Maximum heart rate during YYR2 higher in NSHS than SHS at 36h[a] No differences between conditions for perceptual "overall recovery" or "overall stress. |

*(Continued)*

**Table 1.** (Continued)

| Study | Population and level N (male); age ± SD | Intervention description | Group: Intervention and Control characteristics | Primary Outcome | Secondary outcomes | Results, conclusions and intervention effect |
|---|---|---|---|---|---|---|
| *Marqués-Jiménez et al.* [72]. | 18 semi-professional soccer players 24.0 ± 4.07 years | Evaluate physiological and physical responses to wearing compression garments during soccer matches and during recovery | During game and during 3 days after for 7 h/day. SG: 20–25 mmHg ankle / 15–20 mmHg calf FLG: 25–30 mmHg calf / 15–20 mmHg thigh QG: 15–20 mmHg thigh CG: no compression garments | CMJ (cm) 10–20 m sprint (sec) T-Test (sec) YYIR2 (m) | [La-] mmol/L SaO$_2$ (%) RPE TQR | There are significant correlations, immediately post-match, between 10-m sprint and 20-m sprint in the CG, 10-m sprint and 20-m sprint and 10-m sprint and T-Test in the SG, and [La-] and 10-m sprint in the QG. At 48 h post-match, there are significant correlations between 10-m sprint and 20-m sprint in the EG, 10-m sprint and 20-m sprint in the SG, 10-m sprint and 20-m sprint in the FLG. At 72 h post-match there are significant correlations between 10-m sprint and 20-m sprint in the CG. |
| *Marqués-Jiménez et al.* [73]. | 18 semi-professional soccer players 24.0 ± 4.07 years | Evaluate the influence of different types of compression garments in reducing exercise-induced muscle damage (EIMD) during recovery after a friendly soccer match | During game and during 3 days after for 7 h/day. SG: 20–25 mmHg ankle / 15–20 mmHg calf FLG: 25–30 mmHg calf / 15–20 mmHg thigh QG: 15–20 mmHg thigh CG: no compression garments | | EIMD biomarkers DOMS Swelling | In CG, most biomarkers, including CK, LDH, GOT and GPT, were greater at 72-h post-match compared to pre-match. In EG, increases[a] between pre- and 72-h post-match were observed only in CK and LDH. Thigh swelling increases[a] with time were present in CG. Differences in calf swelling were observed between CG, EG, SG and FLG[a] DOMS differences between groups were only observed between CG, SG and QG in tibialis soreness, between CG and FLG in quadriceps soreness, between CG, EG, SG and QG in calf soreness and between SG and QG in hamstring soreness |

IG: intervention Group; CG: control group; EG: Experimental group; CWI: cold water immersion; TWI: thermoneutral water immersion; SJ: squat jump; CMJ: counter movement jump; MVC: maximal voluntary contraction; DOMS: delayed onset muscle soreness; CK: creatine kinase; Mb: myoglobin; CRP: C-reactive protein; PCM: cooled phase change material; MIVC: maximal isometric voluntary contraction; BAM+: brief assessment of mood; MS: muscle soreness; BFQ: belief questionnaire; SHS: sleep hygiene strategy; NSHS: normal post-game sleep hygiene strategy; YYIR2: Yo-Yo intermittent recovery level 2; RPE: rate perceived exertion; SG: stockings group; FLG: tights group; QG: shorts group; TQR: perceived recovery; [La-]: lactate concentration; SaO$_2$ (%): Arterial oxygen saturation of hemoglobin; EIMD: exercise-induced muscle damage; LDH: lactate; GOT: glutamic oxaloacetic; GPT: glutamic pyruvic

[a] Significance at p<0.05

[b]Significant differences at baseline level (p<0.05)

[70, 72, 73], one assessed the effects of cold-water immersion [57] and one assessed the effects of an acute sleep hygiene strategy [71] on performance outcomes. One of the compression garments interventions [70] combined compression with cold, using specific garments with cooled phase changed material (PCM) at 15º. All the studies assessed the effects of recovery strategies at 24 hours and 48 hours post-match. Since only one of the authors [72, 73] reported the effects of recovery strategies at 72 hours, those values could not be included in the analyses.

Some authors were contacted to provide extra information about the studies. Data from three authors could be obtained [70, 72, 73], two were extracted from the tables and graphs

**Table 2. Risk of bias (RCTs).**

| Study | Domain | | | | | | |
|---|---|---|---|---|---|---|---|
| | Random sequence generation | Allocation concealment | Blinding of participants and researchers | Blinding of outcome assessment | Incomplete outcome data | Selective reporting | Other bias |
| *Ascensão et al.* [57]. | Low | Unclear | High | Unclear | Low | Low | - |
| *Clifford et al.*[70]. | Low | Low | Low | Unclear | Low | Low | Low |
| *Fullagar et al.* [71]. | Low | Unclear | High | Unclear | Low | Low | Low |
| *Marqués-Jiménez et al.* [72]. | Low | Unclear | High | Low | Low | Low | Low |
| *Marqués- Jiménez et al.* [73]. | Low | Unclear | High | Unclear | Low | Low | Low |

[57, 71]. Results of the RCTs risk of bias assessment are presented in Table 2 and Fig 2. The primary source of bias was the blinding of participants and outcome assessors.

## Total estimate

**Primary analyses.** Four RCTs [57, 70–72] were included in the primary analyses for primary outcomes. In total, six analyses were performed: two for CMJ (24h and 48h), two for the 20-m sprint (24h and 48h), and two for MVC (24h and 48h), are shown in Table 3 and Fig 3.

For the CMJ, the results showed no difference at 24h (MD = 1.26; 95% CI: -0.92 to 3.44; p = 0.2575; $I^2$ = 0.0%; SMD = 0.14; 95% CI: -0.31 to 0.59), but greater CMJ values at 48h for the intervention group (MD = 3.01; 95% CI: 1.21 to 4.80; p = 0.001; $I^2$ = 10.4%; SMD = 0.69; 95% CI: 0.14 to 1.25). For the 20-m sprint, the results showed no difference either at 24h (MD = -0.05; 95% CI: -0.14 to 0.04; p = 0.311; $I^2$ = 0%; SMD = -0.28; 95% CI: -0.81 to 0.24), or 48h (MD = -0.02; 95% CI: -0.10 to 0.06; p = 0.592; $I^2$ = 28.1%; SMD = -0.21; 95% CI: -0.74 to 0.31). For the MVC, the results showed no difference either at 24h (MD = -105.41; 95% CI: -189.14 to 399.97; p = 0.483; $I^2$ = 90.4%; SMD = 0.57; 95% CI: -1.10 to 2.25), or 48h for the intervention group (MD = 36.21; 95% CI: -42.58 to 115.01; p = 0.3677; $I^2$ = 0%; SMD = 0.23; 95% CI: -0.38 to 0.84). No heterogeneity was present ($I^2$ range from 0 to 28.1%) in all the analyses, except for MVC at 24h ($I^2$ = 90.4%). Finally, analyses on aerobic capacity (YYIR2) could not be performed due to lack of available data.

**Secondary analyses.** Three RCTs [57, 71, 73] were included in the secondary analyses for the secondary outcomes (physiological markers and wellness data). In total, nine analyses were performed: one for CK, two for CRP (at 24h and 48h), two for quadriceps (QUAD), hamstrings (HAMS), and calf (CALF) DOMS (at 24h and 48h) are shown in Table 3.

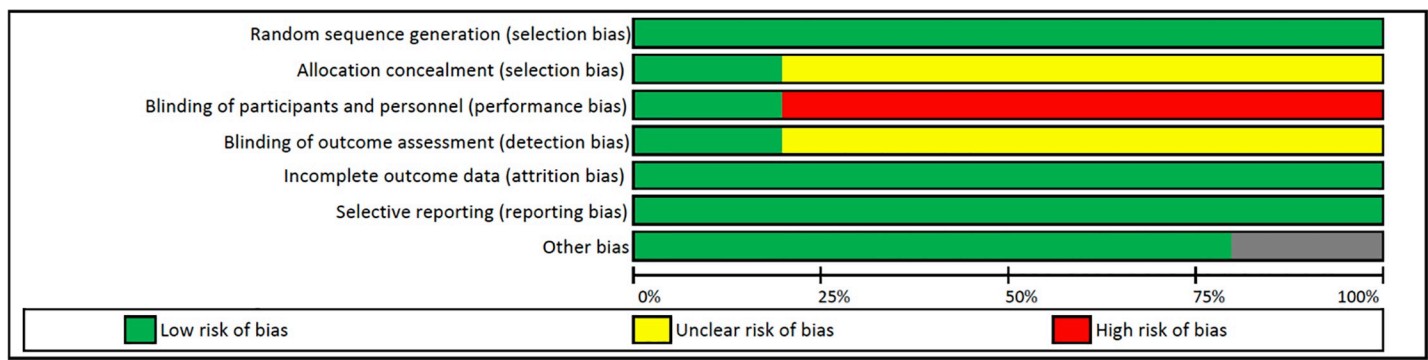

**Fig 2. Risk of bias (RCTs).**

**Table 3. Results from primary and secondary analyses.**

| Variable | | Study | Mean Difference [95% CI] | Random effects model | P-value | SMD [95% CI] |
|---|---|---|---|---|---|---|
| **Primary outcomes** | CMJ 24h | Ascensão et al. 2011 | 4.40 [-1.65; 10.45] | 1.26 [-0.92; 3.44] | 0.2575 | 0.14 [-0.31;0.59] |
| | | Marqués-Jiménez (a) et al. 2018 | 1.04 [-2.84; 4.92] | | | |
| | | Clifford et al. 2018 | 0.65 [-2.27; 3.57] | | | |
| | CMJ 48 h | Ascensão et al. 2011 | 5.90 [1.54; 10.25] | 3.01 [1.21; 4.80] | 0.001 | 0.69 [0.14; 1.25] |
| | | Marqués-Jiménez (a) et al. 2018 | 1.45 [-2.50; 5.40] | | | |
| | | Clifford et al. 2018 | 3.12 [0.99; 5.25] | | | |
| | 20-m sprint 24 h | Ascensão et al. 2011 | -0.40 [-0.19; 0.04] | -0.05 [-0.14; 0.04] | 0.311 | -0.28 [-0.81; 0.24] |
| | | Marqués-Jiménez (a) et al. 2018 | -0.05 [-0.15; 0.05] | | | |
| | 20-m sprint 48 h | Ascensão et al. 2011 | -0.13 [-0.31; 0.05] | -0.02 [-0.10; 0.06] | 0.592 | -0.21 [-0.74; 0.31] |
| | | Marqués-Jiménez (a) et al. 2018 | -0.01 [-0.09; 0.07] | | | |
| | MVC 24 h | Ascensão et al. 2011 | 251.00 [145.62; 356.37] | -105.41 [-189.14; 399.97] | 0.483 | 0.57 [-1.10; 2.25] |
| | | Clifford et al. 2018 | -49.73 [-198.54; 99.08] | | | |
| | MVC 48 h | Ascensão et al. 2011 | 37.00 [-97.17; 171.17] | 36.21 [-42.58; 115.01] | 0.3677 | 0.23 [-0.38; 0.84] |
| | | Clifford et al. 2018 | 34.46 [-107.02; 175.94] | | | |
| **Secondary outcomes** | QS DOMS 24 h | Ascensão et al. 2011 | -2.39 [-3.45; -1.32] | -2.37 [-3.51; -1.22] | <0.0001 | -1.08 [-1.69; -0.48] |
| | | Marqués-Jiménez (b) et al. 2018 | -2.34 [-4.11; -0.57] | | | |
| | QS DOMS 48 h | Ascensão et al. 2011 | -1.56 [-2.46; -0.66] | -1.66 [-2.73; -0.59] | 0.0024 | -0.85 [-1.40; -0.30] |
| | | Marqués-Jiménez (b) et al. 2018 | -1.91 [-3.90; 0.08] | | | |
| | HS DOMS 24 h | Ascensão et al. 2011 | -2.42 [-3.40; -1.45] | -1.95 [-3.17; 0.66] | 0.169 | -0.54 [-2.45; 1.37] |
| | | Marqués-Jiménez (b) et al. 2018 | -0.07 [-2.81; 2.66] | | | |
| | HS DOMS 48 h | Ascensão et al. 2011 | -1.60 [-2.26; -0.93] | -1.46 [-2.34; -0.59] | <0.0001 | -0.50 [-0.89; -0.11] |
| | | Marqués-Jiménez (b) et al. 2018 | -0.61 [-2.98; 1.76] | | | |
| | CS DOMS 24 h | Ascensão et al. 2011 | -3.29 [-3.97; -2.60] | -3.20 [-4.06; -2.35] | <0.0001 | -1.86 [-3.27; -0.45] |
| | | Marqués-Jiménez (b) et al. 2018 | -2.87 [-4.72; -1.01] | | | |
| | CS DOMS 48 h | Ascensão et al. 2011 | 0.89 [0.19; 1.60] | -0.45 [-3.37; 2.48] | 0.7619 | -0.03 [-1.54; 1.50] |
| | | Marqués-Jiménez (b) et al. 2018 | -2.11 [-4.26; 0.04] | | | |
| | CK 24 h[a] | Ascensão et al. 2011 | -168.00 [-225.21; -110.78] | -165.82 [-222.81; -108.83] | <0.0001 | -0.59 [-1.13; -0.08] |
| | | Fullagar et al. 2016 | 112.00 [-534.71; 758.71] | | | |
| | CK 48 h[a] | Ascensão et al. 2011 | -96.00 [-47.60; -44.40] | -93.97 [-145.30; -42.64] | 0.0003 | -0.56 [-1.10; -0.03] |
| | | Fullagar et al. 2016 | 101.00 [-405.11; 607.11] | | | |
| | CRP 24 h[a] | Ascensão et al. 2011 | -0.23 [-0.39; -0.06] | -0.22 [-0.38; -0.06] | 0.0084 | -0.72 [-1.19; -0.24] |
| | | Fullagar et al. 2016 | 0.10 [-0.91; 1.11] | | | |
| | CRP 48 h[a] | Ascensão et al. 2011 | -0.20 [-0.36; -0.05] | -0.21 [-0.35; -0.05] | 0.01 | -0.69 [-1.23; -0.15] |
| | | Fullagar et al. 2016 | -0.60 [-2.40; 1.19] | | | |

SMD: Standardized Mean Difference; CMJ: counter movement jump; MVC: maximal voluntary contraction; DOMS: delayed onset muscle soreness; CK: creatine kinase; CRP: C-reactive protein; QS: quadriceps; HS: hamstrings; CS: calf.

[a]Ascensão evaluated at 24-hours and 48-hours, while Fullagar evaluated at 20-hours and 44-hours post-match.

For the QUAD DOMS, the results showed greater values for the intervention group both at 24h (MD = -2.37; 95% CI: -3.51 to -1.22; $p < 0.0001$; $I^2$ = 0.0%; SMD = -1.08; 95% CI: -1.69 to -0.48) and 48h (MD = -1.66; 95% CI: -2.73 to -0.59; p = 0.0024; $I^2$ = 0.0%; SMD = -0.85; 95% CI: -1.40 to -0.30). For the HAMS DOMS, the results showed no difference at 24h (MD = -1.95; 95% CI: -3.17 to 0.66; p = 0.169; $I^2$ = 55.9%; SMD = -0.54; 95% CI: -2.45 to 1.37), but greater values for the intervention group at 48h (MD = -1.46; 95% CI: -2.34 to -0.59; $p < 0.0001$; $I^2$ = 0.0%; SMD = -0.50; 95% CI: -0.89 to -0.11). For the CALF DOMS, the results only showed greater values for the intervention group at 24h (MD = -3.20; 95% CI: -4.06 to -2.35; $p < 0.0001$;

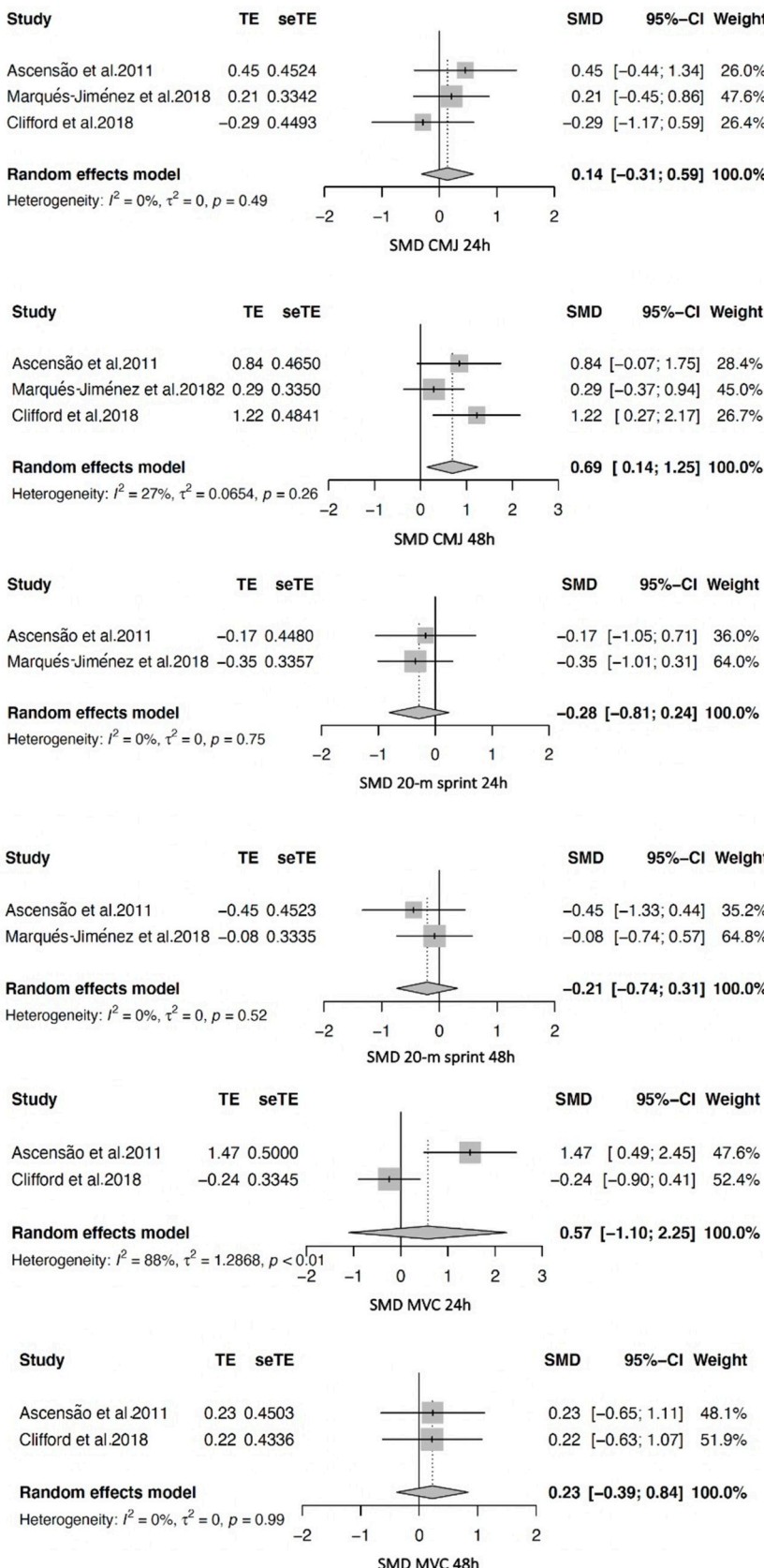

**Fig 3. Meta-analysis of primary outcomes (counter movement jump; 20-m sprint and maximal voluntary contraction) at 24 hours and 48 hours.**

$I^2$ = 0.0%; SMD = -1.86; 95% CI: -3.27 to -0.45), but no difference at 48h (MD = -0.45; 95% CI: -3.37 to 2.48; p = 0.7619; $I^2$ = 83.7%; SMD = -0.03; 95%CI: -1.54 to 1.50). For the CK variables, the results showed greater values for the intervention group both at 24h (MD = -165.82; 95% CI: -222.81 to -108.83; p<0.0001; $I^2$ = 0.0%; SMD = -0.59; 95%CI: -1.13 to -0.08) and 48h (MD = -93.97; 95% CI: -145.30 to -42.64; p = 0.0003; $I^2$ = 0.0%; SMD = -0.56 CI: -1.10 to -0.03). For the CRP, the results showed greater values for the intervention group both at 24h (MD = -0.22; 95% CI: -0.38 to -0.06; p = 0.0084; $I^2$ = 0.0%; SMD = -0.72 CI: -1.19 to -0.24) and 48h (MD = -0.21; 95% CI: -0.35 to -0.05; p = 0.01; $I^2$ = 0.0%; SMD = -0.69; 95% CI: -1.23 to -0.15).

**Post-hoc analyses.** On primary outcomes (Fig 4 and S1 and S2 Tables), neither the intervention nor the control group showed changes in CMJ or 20-m sprint performance at 24h nor 48h compared to baseline, with a trend of decreased performance in both groups. MVC is decreased at 24h and 48h for the control group, and a trend of decreased performance in the intervention group was present.

On secondary outcomes (Fig 4 and S1 and S2 Tables), the intervention group showed no changes for QUAD (24h and 48h), HAMS DOMS (24h and 48h), and CALF DOMS (24h), but decreased CALF DOMS (48h) compared to post-match values. Instead, the control group showed no changes for QUAD DOMS (48h) and HAMS DOMS (24h and 48h), increased QUAD DOMS (24h), and CALF DOMS (24h), but decreased CALF DOMS (48h) compared to post-match values. For CK, both groups showed increased muscle damage at 24h and 48h

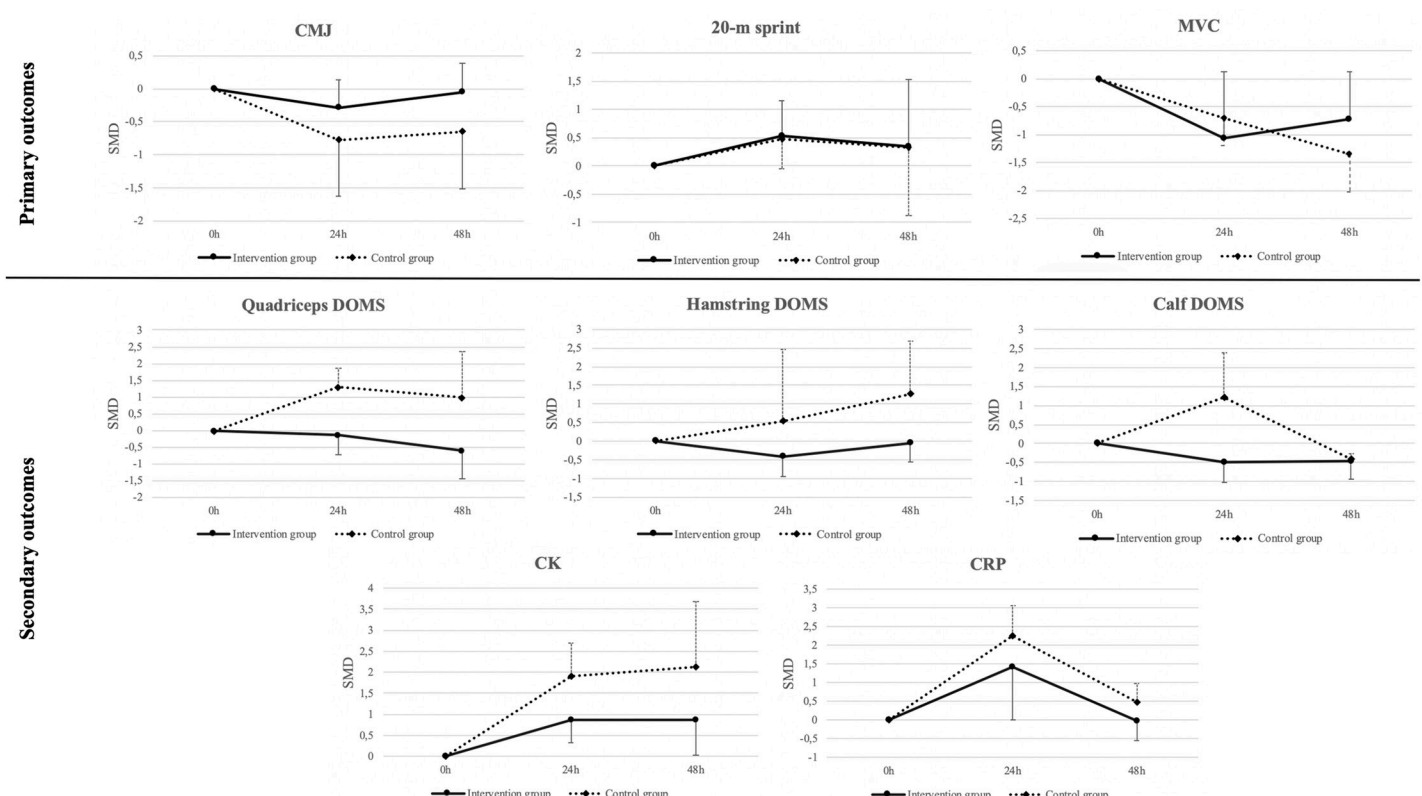

**Fig 4. Time trends for primary and secondary outcomes from the experimental and control group.**

compared to baseline. For CRP, both groups showed increased muscle damage at 24h, but no difference at 48h compared to baseline.

## Discussion

In this systematic review and meta-analyses, where the primary aim was to determine the effects of recovery strategies on post-match performance outcomes, these only provided larger effects on jumping performance at 48h compared to the control group.

### Primary outcome: Jump performance, sprint, and muscle strength

**Between groups.** Our study reveals that for jump performance (CMJ), no significant differences are present in the RCT's at 24h. However, at 48h, there exists a moderate difference (SMD = 0.69) in favor of the intervention group using compression garments or cold-water immersion [57, 70, 72]. This outcome is contrary to that of Rey, Lago-Peñas, Casáis, & Lago-Ballesteros [54] who found that CMJ values in professional soccer players were significantly higher 24h after using an active recovery strategy (12 minutes of submaximal running and 8 minutes of static stretching) after a training session, simulating the demands of a soccer game. The study mentioned earlier could not be included in our meta-analysis due to limitations in its design (performed after training sessions; does not evaluate post-competition recovery effects). However, their findings show that the differences in the benefits arising from the use of single-method recovery strategies and their combinations could be notorious, and should be considered, and more so as was pointed by other authors in their investigations [63].

On the other hand, when analyzing the 20-m sprint and muscle strength (MVC) outcomes, small to moderate non-significant effects are present to enhance recovery when using cold-water immersion and compression garments, at 24h for sprint performance (SMD = -0.28) and after 48h for MVC (SMD = -0.21). Not reaching statistical significance, we cannot confirm that these recovery strategies (cold-water immersion and compression garments) may provide positive sprint performance changes. However, given that a 0.05s difference in 20-m sprint is a meaningful change [74], this could be a relevant trend for future research that can more accurately determine that these strategies have positive effects. These findings do not align with previous studies such as De Nardi, Torre, Barassi, Ricci, & Banfi, [58] or Rowsell, Coutts, Reaburn, & Hill-Haas [55] that found cold-water immersion not to affect physical performance tests. However, the mentioned studies had young players in their samples, and this fact may play an essential role in the differences found between them and those included in our analyzes. This lower effect may exist because young athletes recover faster than adults from strenuous exercise mainly due to lower relative power capabilities, relatively larger flexibility, and enhanced muscle compliance, making them less susceptible to muscle damage [75].

**Within groups.** When looking at time trends, similar trends are present for CMJ and 20-m sprint performance in both the experimental and control group, which tend to decrease at 24h. Then, CMJ seems to remain altered at 48h in the control group but restoring baseline levels in the intervention group. Instead, the 20m-sprint performance seems to remain altered at 48h in both groups. MVC is negatively affected at 24h (SMD = -0.70) and 48h (SMD = -1.34) for the control group. However, for the intervention group, MVC was not affected at any time-point but seemed to follow a similar negative trend (decreased performance). This finding agrees with Thomas, Dent, Howatson & Goodall [76] that found unsolved decrements in MVC in fifteen semiprofessional players 72 hours after a simulated soccer game. The present findings of the within-group analysis need to be taken in caution due to the wide confidence intervals of the outcomes, probably due to the small sample from the analyses.

## Secondary outcome: Psychological, wellness and muscle damage

**Between groups.**   Our study reveals medium to large effects (SMD = -0.50 to -1.86) in favor to the intervention group when analyzing DOMS at 24h and 48h in all muscle groups (quadriceps, hamstrings and calf), except for hamstring at 24h, and calf at 48h, where a similar trend occurs.

For CK and CRP, medium effects (SMD = -0.56 to -0.72) were found in favor of the intervention group at 24h and 48h. This effect may be provided by the reduction in histological damage shown by some studies using different recovery methods [61], or in perceived muscle soreness argued by some others [53].

**Within groups.**   When looking at time trends for each group, an interesting finding arises. The intervention group shows no differences in DOMS perception at 24h and 48h, for any muscle group compared to post-match values but improved DOMS perception in CALF DOMS at 48h. Surprisingly, the intervention group had greater feelings on DOMS after the intervention than in the baseline. However, for the control group, the most significant difference for all muscle groups is at 24h, when an increase in DOMS perception at the QUAD and CALF (SMD = 1.20 to 1.29) exists, and there was no difference for QUAD and HAMS compared to baseline at 48h. Moreover, CALF DOMS showed an improvement compared to baseline (SMD = -0.40) when baseline values were reestablished at 48h, showing that time trends on DOMS are different depending on the muscle group analyzed, especially for QUAD and HAMS which better responses seem to be induced after using recovery strategies. Considering many professional teams compete twice a week, improving the recovery perception may help overall team performance.

When analyzing muscle damage, there is an increase in damage for CK and CRP at 24h compared to baseline in both groups. Moreover, after 48h, CRP returns to the baseline values while CK keeps elevated in both groups. This finding agrees with some authors [32], stating that these biomarkers are sensitive to recovery time.

Although many coaches and practitioners are trying to design and implement protocols based on scientific evidence, we cannot forget the perception of the players regarding this matter. Some studies clearly show that athletes' preferences and scientific evidence do not always agree [77], with perceptual recovery not matching the recovery of performance variables [78]. More than likely, this effect occurs because when professional athletes use recovery strategies, not only are we promoting physical and physiological changes in their bodies, but we are also influencing perceptions and favoring mental well-being from a psychological point of view [53, 60].

## Limitations of the study

One of the most important limitations of this study is the lack of research available in the literature meeting the inclusion criteria. The difficulty of implementing RCT's in team sport environments with competitive settings, truly significant in professional sports, hinders the possibility of providing more evidence to our findings. This gap is significant, and a critical constraint to establishing evidence-based recovery protocols in professional soccer.

Another substantial handicap in our proposal is related to the time frames used by some of the studies included in the sample. The majority of studies used times equal to one day (24h) and two days (48h) for the evaluation of acute recovery. However, some studies used timeframes of 20 and 44 hours that had to be assimilated for performing all our analyses.

Our research group requested all the original datasets to the corresponding authors of the studies included in our final analyses.

## Conclusions

This systematic review and meta-analysis demonstrated that the use of recovery strategies in soccer players such as compression garments, cold water immersion, and sleep hygiene strategy offers greater positive effects only on one of the physical performance tests (CMJ), but no effects on the 20-m sprint or MVC compared to a control group. On top of that, these recovery strategies offer greater positive effects on muscle damage (physiological markers and wellness data) compared to a control group.

The conclusion is based upon the currently available literature where only five RCTs qualified for meta-analysis. We encourage professional practitioners and medical or technical staff teams to implement new RCTs in order to increase current evidence, helping the understanding on how the different recovery interventions and strategies affect physical, physiological and wellness parameters. Another relevant field for future research should aim at investigating the use of recovery strategies specifically by professional teams, as these studies are scarce. Additionally, new protocols based on how these strategies interact, assessing their effectiveness if they are used combined should be implemented and evaluated using the scientific method.

## Supporting information

**S1 File. PRISMA checklist.**
(PDF)

**S2 File. Search strategy.**
(PDF)

**S3 File. Data extraction form for randomised controlled trials.**
(PDF)

**S1 Table. Primary and secondary outcomes for the experimental group relative to the baseline data.** SMD: Standardized Mean Difference; CMJ: counter movement jump; MVC: maximal voluntary contraction; DOMS: delayed onset muscle soreness; CK: creatine kinase; CRP: C-reactive protein; QS: quadriceps; HS: hamstrings; CS: calf. [a]Ascensão evaluated at 24-hours and 48-hours, while Fullagar evaluated at 20-hours and 44-hours post-match.
(DOCX)

**S2 Table. Primary and secondary outcomes for the control group relative to baseline data.** SMD: Standardized Mean Difference; CMJ: counter movement jump; MVC: maximal voluntary contraction; DOMS: delayed onset muscle soreness; CK: creatine kinase; CRP: C-reactive protein; QS: quadriceps; HS: hamstrings; CS: calf. [a]Ascensão evaluated at 24-hours and 48-hours, while Fullagar evaluated at 20-hours and 44-hours post-match.
(DOCX)

## Acknowledgments

The authors would like to give explicit thanks to the authors of the articles included in this systematic review and meta-analysis that provided data used in the original analyses. Also, to Dr. Ibrahim Akubat of Newman University, Birmingham, UK, for his help in the revision and preparation of this manuscript.

## Author Contributions

**Conceptualization:** Albert Altarriba-Bartes, Javier Peña, Jordi Vicens-Bordas, Julio Calleja-González.

**Data curation:** Albert Altarriba-Bartes, Javier Peña, Jordi Vicens-Bordas, Raimon Milà-Villaroel.

**Formal analysis:** Albert Altarriba-Bartes, Javier Peña, Jordi Vicens-Bordas, Raimon Milà-Villaroel, Julio Calleja-González.

**Investigation:** Albert Altarriba-Bartes, Javier Peña, Jordi Vicens-Bordas, Julio Calleja-González.

**Methodology:** Albert Altarriba-Bartes, Javier Peña, Jordi Vicens-Bordas, Raimon Milà-Villaroel, Julio Calleja-González.

**Software:** Raimon Milà-Villaroel.

**Supervision:** Albert Altarriba-Bartes, Javier Peña, Jordi Vicens-Bordas, Julio Calleja-González.

**Validation:** Albert Altarriba-Bartes, Javier Peña, Jordi Vicens-Bordas, Julio Calleja-González.

**Visualization:** Albert Altarriba-Bartes, Javier Peña, Jordi Vicens-Bordas.

**Writing – original draft:** Albert Altarriba-Bartes, Javier Peña, Jordi Vicens-Bordas, Raimon Milà-Villaroel, Julio Calleja-González.

**Writing – review & editing:** Albert Altarriba-Bartes, Javier Peña, Jordi Vicens-Bordas, Julio Calleja-González.

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
