## [Decision Letter · Decision Letter 0]

3 Aug 2020

PONE-D-20-15127

Post-competition recovery strategies in elite male soccer players. Effects on performance: a systematic review and meta-analysis

PLOS ONE

Dear Dr. Peña,

Thank you for submitting your manuscript to PLOS ONE. After careful consideration, we feel that it has merit but does not fully meet PLOS ONE’s publication criteria as it currently stands. Therefore, we invite you to submit a revised version of the manuscript that addresses the points raised during the review process.

Please, check if all points raised by reviewers will be attended.

We look forward to receiving your revised manuscript.

Kind regards,

Moacir Marocolo, Ph.D.

Academic Editor

PLOS ONE

Journal Requirements:

2.Thank you for stating the following financial disclosure:

 [The funders had no role in study design, data collection and analysis, decision to publish, or preparation of the manuscript.].

Reviewers' comments:

Reviewer's Responses to Questions

**Comments to the Author**

1. Is the manuscript technically sound, and do the data support the conclusions?

Reviewer #1: Yes

Reviewer #2: Partly

2. Has the statistical analysis been performed appropriately and rigorously? 

Reviewer #1: I Don't Know

Reviewer #2: Yes

3. Have the authors made all data underlying the findings in their manuscript fully available?

Reviewer #1: Yes

Reviewer #2: No

4. Is the manuscript presented in an intelligible fashion and written in standard English?

Reviewer #1: Yes

Reviewer #2: Yes

5. Review Comments to the Author

Reviewer #1: I believe this is a much needed study to clarify performance and recovery in sport. The suggestions for future studies are well received and your clarification on the discrepancy in the results among various research findings is useful for the researcher.

Reviewer #2: Firstly, I would like to thank the editor of PlosOne for giving me the opportunity to review this manuscript. This SR and MA focuses on recovery strategies in football, which is essential in the area of sport and exercise science and medicine. However, the main weakness of the review is the low number of studies included. This can bias the results and lead practitioners to believe in solutions that are not so reliable and effective to recovery the players. The manuscript is relatively well written, despite a few minor omissions.

Introduction: Generally, this section is well written with logical connection to purpose of the study.

Page 4, Lines 77-83: Please integrate the following key reference related to acute and residual fatigue in soccer.

Silva, J. R., Rumpf, M. C., Hertzog, M., Castagna, C., Farooq, A., Girard, O., & Hader, K. (2018). Acute and residual soccer match-related fatigue: a systematic review and meta-analysis. Sports Medicine, 48(3), 539-583.

Page 5, Lines 104-106: Why do you use an unspecific reference from basketball at this point of the Introduction section? Please add in this paragraph the following recent review about recovery methods in soccer. In the cited review, a suggested time line for recovery methods in soccer has been defined based on scientific evidence and popularity in soccer.

Rey, E., Padrón-Cabo, A., Barcala-Furelos, R., Casamichana, D., & Romo-Pérez, V. (2018). Practical Active and Passive Recovery Strategies for Soccer Players. Strength & Conditioning Journal, 40(3), 40–57.

Page 5, Lines 117-120: At this point of the Introduction, the following key references about the effectiveness of active recovery interventions could be of interest for readers. Please add these references accordingly.

Van Hooren, B., & Peake, J. M. (2018). Do we need a cool-down after exercise? A narrative review of the psychophysiological effects and the effects on performance, injuries and the long-term adaptive response. Sports Medicine, 48(7), 1575-1595.

Ortiz Jr, R. O., Elder, A. J. S., Elder, C. L., & Dawes, J. J. (2019). A systematic review on the effectiveness of active recovery interventions on athletic performance of professional-, collegiate-, and competitive-level adult athletes. The Journal of Strength & Conditioning Research, 33(8), 2275-2287.

Methods

Page 9, Line 224: “(Version 3.5.1.).” instead (Version 3.5.1.)

Page 9, Line 225: “Mean and standarized” instead “Mean, and standarized”

Page 9, Line 239: Is recovery at 48h considered “chronic”? Do you have any reference to support this classification?

Table 1. Why do you include JCR information? You do not only search in WOS database.

Figure 1 and 4 are illegible.

Page 28, Line 367: This is not true. The mentioned study (Rey et al.) has control group. Probably this paper was excluded, as it does not evaluate post-competition effects of recovery means.

Page 28, Lines 367-369: sure? What is the rationale for this statement?

Page 28, Lines 374-375: How can you conclude that CWO and CG may provide positive changes for sprint performance if non-significant effects are present in results?

Page 29, Line 393: “with Thomas et al. [71]”

Page 30, Lines 435-437: Please add references to support this conclusion.

References

First letter of each word on refs title in lowercase letters, please.

6. PLOS authors have the option to publish the peer review history of their article (what does this mean?). If published, this will include your full peer review and any attached files.

Reviewer #1: **Yes: **Ajit Korgaokar

Reviewer #2: No

---

## [Author Response · Author response to Decision Letter 0]

25 Aug 2020

Dear Editor and Reviewers,

We appreciate the time you devoted to reading our manuscript and helping us to craft an improved version providing a thorough review with insightful comments, we have considered all your feedback to improve the final version of the document. We are pleased to clarify your concern, which will enhance the impact and quality of your work. Please find below our responses to your observations. All changes are referenced and visible on the “Manuscript with track changes” file. We have made a concerted attempt to address the specific concerns raised for this systematic review. We have highlighted the changes to this revision for your convenience. 

Review Comments to the Author:

Reviewer #1: I believe this is a much-needed study to clarify performance and recovery in sport. The suggestions for future studies are well received and your clarification on the discrepancy in the results among various research findings is useful for the researcher.

Reviewer #2: Firstly, I would like to thank the editor of PlosOne for giving me the opportunity to review this manuscript. This SR and MA focuses on recovery strategies in football, which is essential in the area of sport and exercise science and medicine. However, the main weakness of the review is the low number of studies included. This can bias the results and lead practitioners to believe in solutions that are not so reliable and effective to recovery the players. The manuscript is relatively well written, despite a few minor omissions.

Authors reply: We appreciate all your comments. As you pointed out, one of the main limitations of the review, already mentioned in the limitations section, is the lack of research available in the literature meeting the inclusion criteria. Given that implementing recovery RCT’s in professional soccer is difficult. Consequently, a low number of studies (although totally updated) could be included. Nevertheless, this manuscript carried out complete literature research, pooling the available results to shed light on the subject and accomplished the main goal to inform the scientific community and helping professional soccer coaches and practitioners to make better decisions and to facilitate the implementation of reliable and effective evidence-based recovery protocols in soccer. 

Introduction: Generally, this section is well written with logical connection to purpose of the study. 

Authors reply: Thanks for your feedback. We appreciate it.

Page 4, Lines 77-83: Please integrate the following key reference related to acute and residual fatigue in soccer.

Silva, J. R., Rumpf, M. C., Hertzog, M., Castagna, C., Farooq, A., Girard, O., & Hader, K. (2018). Acute and residual soccer match-related fatigue: a systematic review and meta-analysis. Sports Medicine, 48(3), 539-583.

Authors reply: Following the reviewer’s suggestion, we have added this reference to add more consistency to the time needed to recover from competitive soccer demands (line 80). Thanks for the comment and for providing us with this reference.

Page 5, Lines 104-106: Why do you use an unspecific reference from basketball at this point of the Introduction section? Please add in this paragraph the following recent review about recovery methods in soccer. In the cited review, a suggested time line for recovery methods in soccer has been defined based on scientific evidence and popularity in soccer. 

Rey, E., Padrón-Cabo, A., Barcala-Furelos, R., Casamichana, D., & Romo-Pérez, V. (2018). Practical Active and Passive Recovery Strategies for Soccer Players. Strength & Conditioning Journal, 40(3), 40–57.

Authors reply: Following the reviewer’s observation, we have eliminated this reference. As pointed out, we used a wrong reference from these group of authors; we have replaced it, lines 104-106, for the correct one:

Calleja-González J, Mielgo-Ayuso J, Sampaio J, Delextrat A, Ostojic SM, Marqués-Jiménez D, et al. Brief ideas about evidence-based recovery in team sports. J Exerc Rehabil. 2018;14: 545–550. doi:10.12965/jer.1836244.122.

Moreover, and following your suggestion we also have added, lines 104-106, the proposed one:

Rey E, Padrón-Cabo A, Barcala-Furelos R, Casamichana D, Romo-Pérez V. Practical Active and Passive Recovery Strategies for Soccer Players. Strength Cond J. 2018;40: 45–57. doi:10.1519/SSC.0000000000000247.

Thanks for the comment and for your interest.

Page 5, Lines 117-120: At this point of the Introduction, the following key references about the effectiveness of active recovery interventions could be of interest for readers. Please add these references accordingly.

Van Hooren, B., & Peake, J. M. (2018). Do we need a cool-down after exercise? A narrative review of the psychophysiological effects and the effects on performance, injuries and the long-term adaptive response. Sports Medicine, 48(7), 1575-1595.

Ortiz Jr, R. O., Elder, A. J. S., Elder, C. L., & Dawes, J. J. (2019). A systematic review on the effectiveness of active recovery interventions on athletic performance of professional-, collegiate-, and competitive-level adult athletes. The Journal of Strength & Conditioning Research, 33(8), 2275-2287.

Authors reply: Following the reviewer’s suggestions, we have added these two references to provide more information on the effectiveness of active recovery interventions, lines 115-117, as follows: 

“Moreover, other authors concluded that even active strategies were largely ineffective for improving post-exercise recovery, offered some benefits compared with passive ones” 

Methods

Page 9, Line 224: “(Version 3.5.1.).” instead (Version 3.5.1.)

Authors reply: Amended (line 224). Thanks.

Page 9, Line 225: “Mean and standarized” instead “Mean, and standarized”

Authors reply: Amended (line 225). Thanks.

Page 9, Line 239: Is recovery at 48h considered “chronic”? Do you have any reference to support this classification?

Authors reply: We agreed with your comments and we have made the change (lines 239-240) to:

 “In the case of studies reporting recovery at different time frames such as 20h and 44 h, those values were assimilated to the ones reported in the literature, 24 h and 48h”. 

Moreover, we have also modified the following sentence, lines 440-441, as we were using words “chronic” and “acute”.

 “This finding agrees with some authors [32], stating that these biomarkers are sensitive to acute and chronic recovery”, for the following one: “This finding agrees with some authors [32], stating that these biomarkers are sensitive to recovery time”, to avoid confusion or misunderstanding. 

Thanks for the comment and for your interest.

Table 1. Why do you include JCR information? You do not only search in WOS database.

Authors reply: Following the reviewer’s observations, we have eliminated Journal of Citation Reports (JCR) information from “Table 1” as we considered that it does not add any relevant information. Thanks for the comment and your interest.

Figure 1 and 4 are illegible.

Authors reply: Thanks for the observation. Although all figures were uploaded according to PLOS requirements, it is possible that accessing them through the PDF file their quality had been affected. However, and due to your comment, we have re-uploaded our figures files to the recommended Preflight Analysis and Conversion Engine (PACE) digital diagnostic tool provided by PLOS.

Page 28, Line 367: This is not true. The mentioned study (Rey et al.) has control group. Probably this paper was excluded, as it does not evaluate post-competition effects of recovery means.

Authors reply: Following the reviewer’s observation, we have eliminated the sentence “absence of an actual control group”. We agreed with the reviewers, and this paper was excluded because it was performed after the training session, so it does not evaluate post-competition recovery effects. We have rewritten the sentence, lines 366-368, as follows: 

“The study mentioned earlier could not be included in our meta-analysis due to limitations in its design (performed after training sessions; does not evaluate post-competition recovery effects)”. 

Thanks for the comment.

Page 28, Lines 367-369: sure? What is the rationale for this statement?

As the reviewer pointed out, we cannot affirm categorically that “…its findings show that the differences in the benefits arising from the use of single-method recovery strategies and their combinations are undoubtedly notorious”. 

We have decided to replace this sentence, lines 369-371, for the following one: 

“…their findings show that the differences in the benefits arising from the use of single-method recovery strategies and their combinations could be notorious and should be considered as pointed by other authors in their investigations [63]”. 

The reference used is the following one: 

García-Concepción MA, Peinado AB, Paredes Hernández V, Alvero-Cruz JR. Efficacy of different recovery strategies in elite football players. Rev Int Med y Ciencias la Act Física y del Deport. 2015;15: 355–389.

Thanks for the comment and for your interest.

Page 28, Lines 374-375: How can you conclude that CWO and CG may provide positive changes for sprint performance if non-significant effects are present in results?

Authors reply: We agree with the reviewer. We made changes, lines 375-379, as follows:

“Not reaching statistical significance, we cannot confirm that these recovery strategies (cold-water immersion and compression garments) may provide positive sprint performance changes. However, given that a 0.05s difference in a 20-m sprint is a meaningful change [74], this could be a relevant trend for future research that can more accurately determine that these strategies have positive effects”.

Page 29, Line 393: “with Thomas et al. [71]”

Authors reply: Amended (lines 405-406). Thanks.

Page 30, Lines 435-437: Please add references to support this conclusion.

As suggested by the reviewer, we have added the following references, lines 449-450, to support this conclusion: 

Rey E, Lago-Peñas C, Lago-Ballesteros J, Casáis L. The Effect of Recovery Strategies on Contractile Properties Using Tensiomyography and Perceived Muscle Soreness in Professional Soccer Players. J Strength Cond Res. 2012;26: 3081–3088. doi:10.1519/JSC.0b013e3182470d3.

Tessitore A, Meeusen R, Cortis C, Capranica L. Effects of different recovery interventions on anaerobic performances following preseason soccer training. J strength Cond Res. 2007;21: 745–50. doi:10.1519/R-20386.1

Thanks for the comment and for your interest.

References. First letter of each word on refs title in lowercase letters, please.

Authors reply: Amended. Thanks.

We hope you find these comments useful. Thanks.

---

## [Decision Letter · Decision Letter 1]

21 Sep 2020

Post-competition recovery strategies in elite male soccer players. Effects on performance: a systematic review and meta-analysis

PONE-D-20-15127R1

Dear Dr. Peña,

We’re pleased to inform you that your manuscript has been judged scientifically suitable for publication and will be formally accepted for publication once it meets all outstanding technical requirements.

Kind regards,

Moacir Marocolo, Ph.D.

Academic Editor

PLOS ONE

Additional Editor Comments (optional):

Reviewers' comments:

Reviewer's Responses to Questions

**Comments to the Author**

1. If the authors have adequately addressed your comments raised in a previous round of review and you feel that this manuscript is now acceptable for publication, you may indicate that here to bypass the “Comments to the Author” section, enter your conflict of interest statement in the “Confidential to Editor” section, and submit your "Accept" recommendation.

Reviewer #2: All comments have been addressed

2. Is the manuscript technically sound, and do the data support the conclusions?

Reviewer #2: Partly

3. Has the statistical analysis been performed appropriately and rigorously? 

Reviewer #2: Yes

4. Have the authors made all data underlying the findings in their manuscript fully available?

Reviewer #2: Yes

5. Is the manuscript presented in an intelligible fashion and written in standard English?

Reviewer #2: Yes

6. Review Comments to the Author

Reviewer #2: Congratulations to authors for their effort. All suggestions were corrected accordingly. Great job.

7. PLOS authors have the option to publish the peer review history of their article (what does this mean?). If published, this will include your full peer review and any attached files.

Reviewer #2: No

---

## [Editor Report · Acceptance letter]

24 Sep 2020

PONE-D-20-15127R1 

Post-competition recovery strategies in elite male soccer players. Effects on performance: a systematic review and meta-analysis 

Dear Dr. Peña:

I'm pleased to inform you that your manuscript has been deemed suitable for publication in PLOS ONE. Congratulations! Your manuscript is now with our production department. 

Kind regards, 

on behalf of

Dr Moacir Marocolo 

Academic Editor

PLOS ONE